# *S100A4* Promotes BCG-Induced Pyroptosis of Macrophages by Activating the NF-κB/NLRP3 Inflammasome Signaling Pathway

**DOI:** 10.3390/ijms241612709

**Published:** 2023-08-11

**Authors:** Mengyuan Li, Yueyang Liu, Xueyi Nie, Boli Ma, Yabo Ma, Yuxin Hou, Yi Yang, Jinrui Xu, Yujiong Wang

**Affiliations:** 1School of Life Sciences, Ningxia University, Yinchuan 750021, China; 20180044@nxmu.edu.cn (M.L.); 18395273708@163.com (Y.L.); 18794898774@163.com (X.N.); 15695013087@163.com (B.M.); myb512816@163.com (Y.M.); 18209689779@163.com (Y.H.); yangyi@nxu.edu.cn (Y.Y.); 2Key Laboratory of Ningxia Minority Medicine Modernization, Ministry of Education, Ningxia Medical University, Yinchuan 750004, China

**Keywords:** *S100A4*, pyroptosis, *Mycobacterium tuberculosis*, NLRP3 inflammasome, NF-κB

## Abstract

Pyroptosis is a host immune strategy to defend against *Mycobacterium tuberculosis* (Mtb) infection. *S100A4*, a calcium-binding protein that plays an important role in promoting cancer progression as well as the pathophysiological development of various non-tumor diseases, has not been explored in Mtb-infected hosts. In this study, transcriptome analysis of the peripheral blood of patients with pulmonary tuberculosis (PTB) revealed that *S100A4* and *GSDMD* were significantly up-regulated in PTB patients’ peripheral blood. Furthermore, there was a positive correlation between the expression of *GSDMD* and *S100A4*. KEGG pathway enrichment analysis showed that differentially expressed genes between PTB patients and healthy controls were significantly related to inflammation, such as the NOD-like receptor signaling pathway and NF-κB signaling pathway. To investigate the regulatory effects of *S100A4* on macrophage pyroptosis, THP-1 macrophages infected with *Bacillus Calmette-Guérin* (BCG) were pre-treated with exogenous *S100A4*, *S100A4* inhibitor or si-*S100A4*. This research study has shown that *S100A4* promotes the pyroptosis of THP-1 macrophages caused by BCG infection and activates NLRP3 inflammasome and NF-κB signaling pathways, which can be inhibited by knockdown or inhibition of *S100A4*. In addition, inhibition of NF-κB or NLRP3 blocks the promotion effect of *S100A4* on BCG-induced pyroptosis of THP-1 macrophages. In conclusion, *S100A4* activates the NF-κB/NLRP3 inflammasome signaling pathway to promote macrophage pyroptosis induced by Mtb infection. These data provide new insights into how *S100A4* affects Mtb-induced macrophage pyroptosis.

## 1. Introduction

Tuberculosis (TB) is a respiratory infectious disease caused by *Mycobacterium tuberculosis* (Mtb) [1]. According to the WHO 2022 Global Tuberculosis Report, tuberculosis is the second most deadly infectious disease after coronavirus disease 2019 (COVID-19), and it ranks as the thirteenth leading cause of death worldwide. At the same time, it is also the leading cause of mortality among individuals infected with HIV and the primary fatally infectious disease associated with antimicrobial resistance. The COVID-19 pandemic has undone years of global progress in the fight against TB and continues to exert a devastating impact on TB control and prevention efforts [2,3]. Despite extensive research on TB and Mtb, their intricate and variable pathogenesis remains incompletely understood. To effectively combat the public health crisis caused by TB, it is crucial to investigate the pathogenesis of TB and discover new therapeutic targets [4].

Alveolar macrophages are the main immune cells involved in Mtb infection and TB progression [5]. When Mtb invades the body, macrophages act as the first line of defense against innate immunity by using pattern recognition receptors (PRRs) on their surface to rapidly recognize the pathogen-associated molecular patterns (PAMPs) of Mtb. Macrophages engulf Mtb into phagosomes and kill it through fusion with lysosomes and acidification of phagolysosomes. As intracellular bacteria, Mtb can inhibit or evade innate and adaptive immunity through a variety of strategies. Macrophages can also defend against Mtb infection by initiating programmed cell death, such as apoptosis and autophagy [6,7,8,9]. In recent years, some research studies have indicated that macrophage pyroptosis plays an extraordinary role in defending against invasion by Mtb [10,11,12].

Pyroptosis is characterized by cellular swelling, plasma membrane rupture, and the release of the cell’s cytoplasmic contents, which promotes an inflammatory response [13]. This process is initiated by the activation of inflammasomes that activate caspase protease, ultimately resulting in lytic cell death mediated by the effector protein Gasdermin D (GSDMD). Canonical inflammasomes comprise a nucleotide oligomerization domain receptor (NLR, NOD-like receptors), an adaptor protein (ASC; Apoptosis-associated speck-like protein containing a CARD (Caspase recruitment domain)) and an effector protease, Caspase-1 [14]. Among the NLR family members, NLRP3 is one of the most critical proteins and also serves as a key PRR that recognizes Mtb infection and initiates the pyroptosis pathway [15]. The NLRP3 inflammasome triggers the activation of caspase-1 and cleavage of pro-IL-1β, pro-IL-18, and GSDMD [16,17]. Upon cleavage, the released N-terminal domain of GSDMD forms a channel in the cell membrane that disrupts the osmotic pressure inside and outside the cell, causing cell distension and lysis. The mature IL-18 and IL-1β are released outside the cell through GSDMD pores, triggering pyroptosis and inflammatory response [18].

The S100 protein family is a group of calcium-binding proteins, which now has 25 members with high sequence and structure similarity [19]. Within cells, S100 proteins regulate cell proliferation, differentiation, migration, energy metabolism, Ca^2+^ homeostasis, inflammation, and cell death [20]. *S100A4* is an important member of the S100 protein family, expressed in various cell types, including lymphoid and myeloid cells such as macrophages, neutrophils, mast cells, and memory T cells. It plays a crucial role in promoting cancer occurrence and metastasis [21,22,23,24]. Research has shown that *S100A4* plays different roles in the pathogenesis of infectious viruses. Yang’s studies show that *S100A4*^+^ macrophages promote the infiltration of ZIKV and the continuous presence of interferon serotonin secreted by the macrophages in the seminiferous tubule, which can increase the permeability of the blood–testis barrier [25]. Anju’s study found that *S100A4* is involved in the process of epithelial–mesenchymal transition (EMT) in human immunodeficiency virus (HIV)-associated nephropathy (HIVAN), and renal cell EMT plays a significant role in the development of proliferative HIVAN phenotype [26]. Zhu reported that increasing the expression of the *S100A4* can lead to the development of liver cancer, which suggests that it could be a potential therapeutic target inducing liver cancer in patients with HBV [27]. These studies showed that *S100A4* is involved in the pathological and physiological processes of viral infectious diseases.

However, the role of *S100A4* in the host response to Mtb infection remains unexplored. In this study, THP-1 macrophages were infected with *Bacillus Calmette-Guérin* (BCG) to explore the regulatory effect of *S100A4* on pyroptosis based on transcriptome analysis of peripheral blood of PTB patients. Our experiments indicate that *S100A4* is a positive regulator of pyroptosis through si-*S100A4*, exogenous *S100A4*, or inhibition of *S100A4* pre-treatment. Moreover, the addition of NF-κB inhibitor Triptolide (TPL) or NLRP3 inhibitor MCC950 prevented the up-regulation of pyroptosis by exogenous *S100A4*. These results indicated that *S100A4* promoted pyroptosis by activating the NF-κB/NLRP3 signaling pathway.

## 2. Results

### 2.1. Transcriptome Analysis of Peripheral Blood from TB Patients

Based on peripheral blood mRNA transcriptome data (GSE83456) from healthy individuals, patients with clinically active pulmonary tuberculosis (PTB), and patients with extrapulmonary tuberculosis (EPTB) provided by the UK National Institute for Medical Research (NIMR, England) [28], the information details can be found in Table 1. This study employed healthy people (N = 61) and PTB patients (N = 45) for data mining analysis; we found that *S100A4* and *GSDMD* were significantly up-regulated in PTB patients compared with healthy controls (*p* < 0.001) (Figure 1A,B). Spearman’s correlation coefficient was used to analyze the correlation between *S100A4* and *GSDMD*. The analysis revealed a positive correlation, r = 0.460 (Figure 1C). The AUC value of the *S100A4* ROC curve was 0.705, while that of the *GSDMD* ROC curve was 0.940 (Figure 1D,E), indicating their high diagnostic value. We conducted differential expression analysis of all genes in both healthy and PTB groups, the volcano plots and histograms were utilized to analyze the expression levels of 1130 differentially expressed genes (DEGs), which consisted of 755 up-regulated genes and 375 down-regulated genes, and the criterion is FC ≥ 1.5, *p* < 0.05 (Figure 2A,B). KEGG pathway enrichment analysis showed that these DEGs were significantly enriched in inflammation-related pathways such as the NOD-like receptor signaling pathway, NF-κB signaling pathway, Toll-like receptor signaling pathway, and TNF signaling pathway (Figure 2C,D).

### 2.2. Infection of THP-1 Macrophage with BCG Up-Regulate S100A4 and Induce Pyroptosis

To investigate the regulatory effect of *S100A4* on macrophages pyroptosis induced by Mtb infection, a Western blot was performed to evaluate the expression of *S100A4* and GSDMD in THP-1 macrophages at different time points (0 h, 2 h, 6 h, 12 h, 24 h, 48 h) after BCG infection. The results showed that BCG infection up-regulated the expression of *S100A4*, which peaked at 24 h (Figure 3A,B). The expression of GSDMD-N was consistent with that of *S100A4* (Figure 3A,C). ELISA results showed that the extracellular concentrations of IL-1β and IL-18 also increased gradually (Figure 3D,E). CCK-8 results indicated a gradual decrease in cell viability with the extension of infection time (Figure 3F). The protein expression of *S100A4* and GSDMD was detected using immunofluorescence staining, and cell morphology was observed using transmission electron microscopy. The results of immunofluorescence staining showed that the BCG group (BCG infection for 24 h) had higher expression levels of *S100A4* and GSDMD compared to the control group (Figure 3G). The results of transmission electron microscopy exhibited that the cell membrane of the control group was intact without perforation (green arrow), while the cell membrane of the BCG group showed holes (red arrow) and increased lipid droplets (yellow arrow) (Figure 3H). These findings suggest that BCG infection enhances the expression of *S100A4* and induces pyroptosis in THP-1 macrophages.

### 2.3. si-S100A4 Inhibited Pyroptosis in BCG-Infected THP-1 Macrophage

To confirm the regulatory effect of *S100A4* on pyroptosis after BCG infection of THP-1 macrophages, we screened for a si*RNA* that could effectively silence the expression of *S100A4* (Figure 4A–C). The results of the Western blots showed that *S100A4* knockdown inhibited the up-regulation of pyroptosis-related proteins GSDMD-N, IL-1β p17, and IL-18 p22 in THP-1 macrophages caused by BCG infection (Figure 4A,D,E). The results of qRT-PCR were consistent with these findings (Figure 4G–I). ELISA results showed that the knockdown of *S100A4* significantly reduced the release of inflammatory factors IL-1β and IL-18 caused by BCG infection (Figure 4J,K). Immunofluorescence staining results demonstrated that pre-treatment with *S100A4* knockdown significantly decreased the expression of both *S100A4* and GSDMD in BCG-infected THP-1 macrophages (Figure 4L). These findings confirm that inhibiting *S100A4* suppresses macrophage pyroptosis induced by BCG infection.

### 2.4. si-S100A4 Inhibited the Activation of NLRP3 Inflammasome in BCG-Infected THP-1 Macrophage

Given that many previous studies have reported that activation of the NLRP3 inflammasome can cause pyroptosis [29], we further explored the regulatory effect of *S100A4* on NLRP3 inflammasome in BCG-infected THP-1 macrophages. Western blot and qRT-PCR results showed that compared with the si-*NC* group, the protein and mRNA expressions of inflammasome-related proteins NLRP3, Caspase-1 p48, and ASC were significantly up-regulated in the BCG group. The protein and mRNA expressions of NLRP3, Caspase-1 p48, and ASC were significantly lower in the si-*S100A4*+BCG group than those in the BCG group (Figure 5A,G). Immunofluorescence staining further confirmed that inhibition of *S100A4* expression reduced NLRP3 expression in BCG-infected THP-1 macrophages (Figure 5H). To explore the regulatory role of NLRP3 in macrophage pyroptosis during BCG infection, we co-treated THP-1 macrophages with the NLRP3 inhibitor MCC950 and BCG. Western blot results showed that the expression of inflammasome-related proteins NLRP3, Caspase-1 p48, and ASC was significantly decreased in BCG-infected THP-1 macrophages after MCC950 treatment. The expression of the pyroptosis-related proteins GSDMD-N, IL-1β, and IL-18 was also significantly reduced (Figure 6A–C).

### 2.5. Exogenous S100A4 Up-Regulated GSDMD-N and NLRP3 Protein Expression, While Niclosamide Had the Opposite Effects in BCG-Infected THP-1 Macrophage

To further confirm the regulatory effect of *S100A4* on pyroptosis of THP-1 macrophages infected with BCG, exogenous *S100A4* and *S100A4* specific inhibitor Niclosamide (Nic) were added prior to treatment. Western blot revealed that the expression of *S100A4* was significantly increased in THP-1 macrophages at a concentration of 1.5 ug/mL (Figure 7A,B). Nic at a concentration of 1 uM effectively reduced the expression level of *S100A4* (Figure 7A,C). Therefore, we used an exogenous concentration of 1.5 ug/mL for *S100A4* and a concentration of 1 uM for Nic. The protein expression levels of GSDMD-N and NLRP3 were significantly up-regulated after the addition of *S100A4* in BCG infection. However, upon adding Nic, the protein expression of GSDMD-N and NLRP3 was significantly down-regulated (Figure 7D–F). These results suggest that *S100A4* promotes pyroptosis in BCG-infected THP-1 macrophages, whereas its inhibition suppresses pyroptosis.

### 2.6. S100A4 Up-Regulates Pyroptosis of BCG-Infected THP-1 Macrophage by Activating the NF-κB/NLRP3 Inflammasome Signaling Pathway

NF-κB signaling is a key transcription factor in the regulation of inflammation and initiates the activation of the NLRP3 inflammasome [30,31,32]. Knockdown of *S100A4* expression by si-*S100A4* inhibited the phosphorylation of NF-κB in BCG-infected THP-1 macrophages (Figure 8A,B). We hypothesized that *S100A4* might regulate the pyroptosis of BCG-infected THP-1 macrophages through the NF-κB signaling pathway.

Triptolide (TPL), one of the major bioactive ingredients in the Chinese traditional Herb *Tripterygium wilfordii Hook f* (TWH f), has been used to treat inflammatory, autoimmune, and malignant diseases for centuries [33]. It is also a specific inhibitor of NF-κB [34,35]; after co-treatment of THP-1 macrophages with TPL and BCG, the Western blot results showed significantly decreased protein expressions of p-NF-κB and NF-κB (Figure 8C–E). The expression levels of inflammasome-related proteins, including NLRP3, Caspase-1 p48, ASC, and pyroptosis-related protein GSDMD-N, were significantly downregulated. These findings suggest that the NF-κB signaling pathway is involved in regulating NLRP3 inflammasome activation and pyroptosis (Figure 8C,F).

This study confirmed that inhibition of *S100A4* could down-regulate the protein expression of GSDMD-N, NLRP3, and NF-κB in BCG-infected THP-1 macrophages. This suggests that *S100A4* promotes BCG-induced pyroptosis of THP-1 macrophages by activating NF-κB/NLRP3 inflammasome signaling pathway. THP-1 macrophages were pr-treated with TPL or MCC950 and infected with BCG. The results of Western blot showed that the addition of TPL and MCC950 inhibited the promotion effect of *S100A4* on pyroptosis during BCG infection (Figure 9A,B), indicating that *S100A4* regulates pyroptosis through NF-κB and NLRP3 inflammasome. Notably, the expression of p-NF-κB did not change significantly after the addition of MCC950, but the expression of NLRP3, Caspase-1, and ASC decreased after the addition of TPL (Figure 9A–F). This indicates that *S100A4* regulates THP-1 macrophage pyroptosis through NF-κB/NLRP3 inflammasome signaling pathway.

## 3. Discussion

About one-third of the global population is infected with Mtb. Despite being a curable and preventable disease, MDR-TB presents a substantial health hazard, particularly in developing nations [36]. As an adaptable intracellular pathogen that coevolves within the host, Mtb has developed numerous strategies, including immune evasion, to establish long-term infection [37]. To develop novel therapies for TB, it is imperative to effectively target the bacterial immune evasion mechanism. The role of pyroptosis in host anti-infection and tumor immunity has gained increasing attention [38,39]. Chai et al. found that GSDMD-mediated pyroptosis and release of inflammatory cytokine play a crucial role in host anti-Mtb infection; GSDMD can provide an early and robust protective immune response against infection, thereby limiting Mtb growth [40]. Our study provided evidence that the expression of the pyroptosis-related protein GSDMD was up-regulated in PTB patients and BCG-infected THP-1 macrophages, suggesting Mtb infection induces pyroptosis. 

*S100A4* plays an important role in immune system regulation, particularly in the modulation of inflammatory responses [41,42] across diverse diseases, including colorectal cancer [43], amyotrophic lateral sclerosis (ALS) [44], and asthma [45]. However, its involvement in pathogen-mediated infections remains largely unexplored. Thus, we aimed to investigate the impact of *S100A4* after Mtb infection. This study confirmed that *S100A4* was up-regulated in PTB patients and BCG-infected THP-1 macrophages. Moreover, *S100A4* promoted BCG-induced pyroptosis of THP-1 macrophages.

Previous studies have demonstrated the involvement of the NF-κB/NLRP3 inflammasome signaling pathway in regulating pyroptosis in various diseases, such as spinal cord injury [46], diabetic cardiomyopathy [47] and *Aspergillus fumigatus* keratitis [48]. In this study, KEGG analysis of differentially expressed genes in TB patients’ peripheral blood transcripts revealed a significant enrichment of the NOD-like receptor (NLR) signaling pathway, NF-κB signaling pathway, and other inflammation-related pathways. This study has also confirmed that the *S100A4* promotes pyroptosis by activating the NF-κB/NLRP3 inflammasome signaling pathway in BCG-infected THP-1 macrophages while inhibiting the expression of NF-κB or NLRP3 down-regulated pyroptosis related proteins (GSDMD-N, IL-1β, IL-18) and NLRP3 inflammasome related proteins (NLRP3, Caspase-1, ASC) in BCG-infected THP-1 macrophages. 

During Mtb infection, the up-regulation of *S100A4* expression in macrophages promotes inflammasome activation and pyroptosis, leading to the formation of a beneficial inflammatory microenvironment for host defense against Mtb infection. However, the persistence of Mtb infection leads to the development of an increasingly excessive or uncontrollable inflammatory microenvironment. Pyroptosis ensues with the release of intracellular live Mtb and their diffusion to surrounding cells, thereby facilitating Mtb spread. This dynamic inflammatory response reflects the delicate balance between protective immunity and immunopathology, thereby enhancing our comprehension of the intricate regulatory mechanisms governing cellular inflammation and pyroptosis signaling pathways during pathogen infection.

Our results were obtained from macrophages infected with an attenuated strain of *Mycobacterium bovis* and did not fully replicate the process of Mtb infection in humans. However, certain experiments have validated that the expression patterns of IL-1β and other pro-inflammatory factors, as well as immunomodulatory genes induced by H37RV, BCG, and H37RA infections in THP-1 macrophages, are consistent, along with the activation patterns of Caspase-1 and Cleaved-Caspase-1 signal transduction pathways [49]. Perhaps the disparity lies in the fact that H37RV infection displays a more copious transcriptional pattern of virulence factors compared to H37RA or BCG. Therefore, our forthcoming research will concentrate on investigating pyroptosis induced by Mtb H37RV and H37RA infections in macrophages and *S100A4*^−/−^ mice.

In summary, during Mtb infection, the up-regulation of *S100A4* expression in macrophages promotes pyroptosis by NF-κB/NLRP3 inflammasome signaling pathway. This study enhances our comprehension of the intricate regulatory mechanisms governing cellular inflammation and pyroptosis signaling pathways during pathogen infection.

## 4. Materials and Methods

### 4.1. Bacterial Culture

BCG was purchased from the Centers for Disease Control and Prevention (CCDC, Beijing, China) and cultured in Middlebrooks 7H9 broth medium (M1315, BD, San Jose, CA, USA) containing 10% oleic acid albumin dextrose catalase (BD Diagnostic Systems, USA) and 0.05% Tween-80 at 37 °C in a 5% CO_2_ tissue culture incubator until it reached the early logarithmic phase of growth.

### 4.2. Cell Culture and Infection

Human monocyte-macrophage THP-1 cells were acquired from the cell bank of the Chinese Academy of Sciences and cultivated in RPMI-1640 containing 10% FBS at 37 °C in a 5% CO_2_ incubator until 80–90% confluent. These cells were then inoculated into plates containing 6 wells (1 × 10^6^ cells/well) with media containing 50 ng/mL Phorbol myristate acetate (PMA) to induce adherence. After being cultured for 48 h, the media were replaced with fresh culture media, and cells were used for infection after an additional 24 h culture period. For bacterial infection, adherent cells were infected with BCG at an MOI of 10.

### 4.3. Small Interfering RNA Transfection

Nontarget control si*RNA* and small interfering RNA sequences against *S100A4* (si-*S100A4*), as listed in Table 2. were designed and generated by Genepharma Co., Ltd. (Shanghai, China). THP-1 macrophages were plated at 1 × 10^6^ cells per well in a 6-well plate the day before transfection. The si*RNAs* were transfected into the cells using Lipofectamine^TM^ RNAi MAX reagent according to the manufacturer’s instructions. After incubating for an additional 24 h at 37 °C in a CO_2_ incubator, the cells were harvested to analyze the expression of genes of interest using Western blot assays and quantitative reverse-transcription PCR (qRT-PCR).

### 4.4. Chemicals and Inhibitors Treatment

NF-κB inhibitor Triptolide (TPL) was purchased from MedChemExpress (HY-32735, Monmouth Junction, NJ, USA), and NLRP3 inhibitor MCC950 was purchased from MedChemExpress (HY-12815, Monmouth Junction, NJ, USA). THP-1 macrophages were pr-treated for 2 h using TPL (5 nM) and MCC950 (10 μM), respectively. Followed by infection with BCG for 24 h. Exogenous *S100A4* was purchased from Sino Biological (10185-H01H, Beijing, China). The *S100A4* inhibitor Niclosamise (Nic) was purchased from MedChemExpress (HY-B0497, Monmouth Junction, NJ, USA). THP-1 macrophages were pr-treated with Exogenous *S100A4* at concentrations of 0, 0.25, 0.50, 1.00, and 1.50 μg/mL, Nic at concentrations of 0, 0.5, 1.0, 3.0, and 5.0 μM for 2 h before being infected with BCG for 24 h, respectively. Cellular protein samples were extracted and stored at −20 °C for Western blot analysis.

### 4.5. Western Blot

Key antibodies used included anti-*S100A4* (16105-1-AP, Proteintech, Wuhan, China); β-actin (20536-1-AP, Proteintech, Wuhan, China); anti-GSDMD-N (39754S, Cell Signaling Technology, Danvers, MA, USA); anti-Caspase1 p48 (3866S, Cell Signaling Technology, Danvers, MA, USA); anti-ASC (13833S, Cell Signaling Technology, Danvers, MA, USA); anti-NLRP3 (15101S, Cell Signaling Technology, Danvers, MA, USA); anti-IL-1β p17 (83186S, Cell Signaling Technology, Danvers, MA, USA); anti-IL-18 (54943S, Cell Signaling Technology, Danvers, MA, USA).

A total protein extraction kit was used to extract protein from cells. The protein levels were then quantified and separated using SDS-PAGE before being transferred onto an appropriate membrane (300 mA, 2 h). The blots were blocked for 1 h at room temperature, followed by overnight probing with antibodies specific for β-actin (1:3000), *S100A4* (1:1000), GSDMD (1:1000), NLRP3 (1:1000), Caspase-1 p48 (1:1000), ASC (1:500), IL-1β (1:1000), IL-18 (1:1000), NF-κB/p-NF-κB (1:1000) at 4 °C. Blots were probed with an HRP-conjugated secondary antibody (1:3000) for 1 h at room temperature. Protein bands were detected using a chemiluminescence kit (Absin, Shanghai, China) and an Amersham Imager 6000. 

### 4.6. Quantitative Reverse-Transcription PCR (qRT-PCR)

Total cellular RNA was extracted using Trizol and then reverse transcribed into cDNA following the instructions of the reverse transcription kit. qRT-PCR was performed using cDNA as a template. The results were analyzed using the 2^−ΔΔCt^ method in triplicates in three different independent experiments. The primers used for qRT-PCR are listed in Table 3.

### 4.7. Immunofluorescent Staining

Cells were cultivated in plates containing 12 wells (1.0 × 10^5^ cells/well) for appropriate periods. After that, the cells were fixed with 4% paraformaldehyde for 30 min, rinsed thrice with PBS, permeabilized using 0.5% Triton-X 100 for 20 min, washed thrice with PBS, and incubated overnight with anti-*S100A4*, anti-GSDMD, and anti-NLRP3. Following three additional washes with PBS, the samples were probed for 1 h using FITC-conjugated AffiniPure Goat Anti-Rabbit IgG (H+L) antibody (SA00001-2, Proteintech, Wuhan, China), rinsed thrice with PBS, and nuclei were counterstained using DAPI. After three additional washes with PBS, the samples were sealed and imaged via an inverted fluorescence microscope (Olympus, Tokyo, Japan).

### 4.8. Transmission Electron Microscopy

THP-1 macrophage samples were fixed in 2.5% glutaraldehyde (pH 7.4) for 2 h, washed three times with 0.1 M phosphate buffer (pH 7.2), and fixed in 1% osmic acid at 4 °C for another 2 h. These samples were subsequently dehydrated using a graded ethanol series before being embedded in Epon–Araldite resin and placed into a model for polymerization. Ultrathin slices (70–80 nm) are treated with the anti-stain uranium acetate and imaged using a Hitachi H-7650 Electron Microscope (Tokyo, Japan).

### 4.9. Enzyme-Linked Immunosorbent Assay (ELISA)

An enzyme-linked immunosorbent assay (ELISA) kit for human IL-1β (Boster, EK0392, Beijing, China) and IL-18 (Boster, EK0864, Beijing, China) was purchased from Boster. Supernatants (500 μL) were collected from each group, and ELISAs were performed according to the instructions provided with the ELISA kit. Cytokine concentrations in individual samples were quantified by measuring absorbance at 450 nm with a microplate reader.

### 4.10. CCK-8 Assay

Cell Counting Kit-8 was purchased from Absin (abs50003, Absin, Shanghai, China). THP-1 macrophages at the logarithmic growth stage were taken and treated with a cell density of 1 × 10^4^·mL^−1^, then inoculated into 96 well culture plates. After culturing cells for 48 h and infecting them with BCG for 24 h, 10 μL of CCK-8 solution was added to each well. The absorbance (OD) of each well was measured at a wavelength of 450 nm using a fluorescence microplate reader, and data were recorded after 4 h. Cell viability (%) = (OD value of experimental group − OD value of blank group)/OD value of control group × 100%.

### 4.11. GEO Data Analysis

The GSE83456 dataset is from the Gene Expression Omnibus (GEO) database, which includes peripheral blood transcriptome sequencing information from 61 healthy controls (controls), 47 patients with extrapulmonary tuberculosis (EPTB), and 45 patients with pulmonary tuberculosis (PTB). Healthy people (N = 61) and PTB patients (N = 45) in these datasets were selected as research objects. These datasets were centralized and normalized using the R language scale function. Then, the expression levels of *GSDMD* and *S100A4* in the healthy group and PTB group were compared, and the *t*-test was used to determine the significance level of difference. The ggpubr package of the R programming language was used to draw box plots and correlation scatter plots, and the pROC package of the R programming language was utilized for conducting ROC analysis and drawing ROC curves. The limma algorithm was employed to analyze differences in all genes between samples from the healthy and PTB groups. FC ≥ 1.5, *p* < 0.05 were the criterion of analysis for DEGs. Volcano plots were plotted using the R package ggplot2. Histograms of gene numbers were plotted using GraphPad Prism 8.0. For the differential expressed genes obtained, the R programming language ClusterProfiler was used for KEGG pathway enrichment analysis, and the top 30 most significantly enriched pathways were screened. Bubble maps were then drawn using the ggpubr package.

### 4.12. Statistical Analysis

GraphPad Prism 8.0 was used for all statistical analyses, and results from triplicate experiments were compared using *t*-tests or one-way ANOVAs. These data were expressed as mean ± standard deviation (SD), and * *p* < 0.05; ** *p* < 0.01; *** *p* < 0.001.

## 5. Conclusions

Our results indicate that *S100A4* is a positive regulator of pyroptosis in BCG-infected macrophages and promotes pyroptosis by activating the NF-κB/NLRP3 inflammasome signaling pathway (Figure 10).

## Figures and Tables

**Figure 1 ijms-24-12709-f001:**
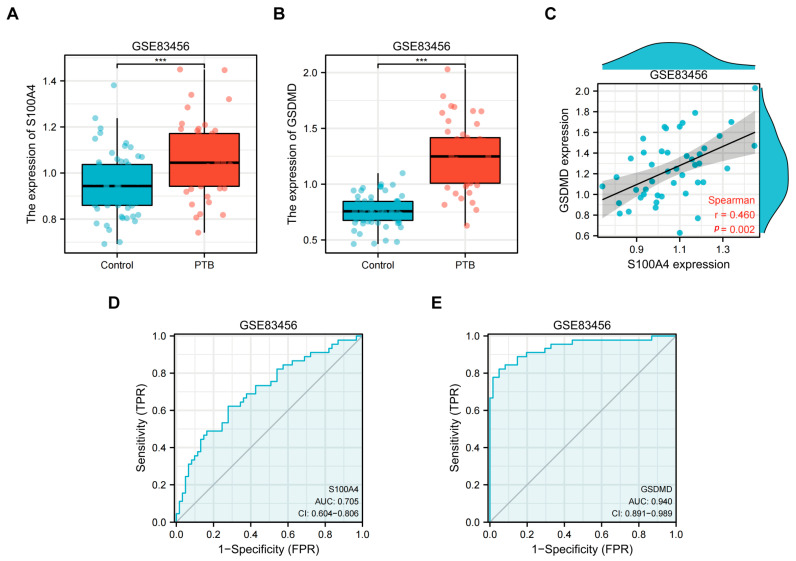
*S100A4* and *GSDMD* were regulated in the peripheral blood of tuberculosis patients. The GEO database GSE83456 (peripheral blood transcriptome sequencing dataset of tuberculosis patients) was used to collect the peripheral blood of healthy individuals (Control, N = 61) and pulmonary tuberculosis patients (PTB, N = 45) for transcriptome analysis. (**A**,**B**) Differential expression of (**A**) *S100A4* and (**B**) *GSDMD* transcripts in peripheral blood of tuberculosis patients. (**C**) Correlation between *GSDMD* and *S100A4*. (**D**,**E**) ROC curves of *S100A4* and *GSDMD*. *** *p* < 0.001.

**Figure 2 ijms-24-12709-f002:**
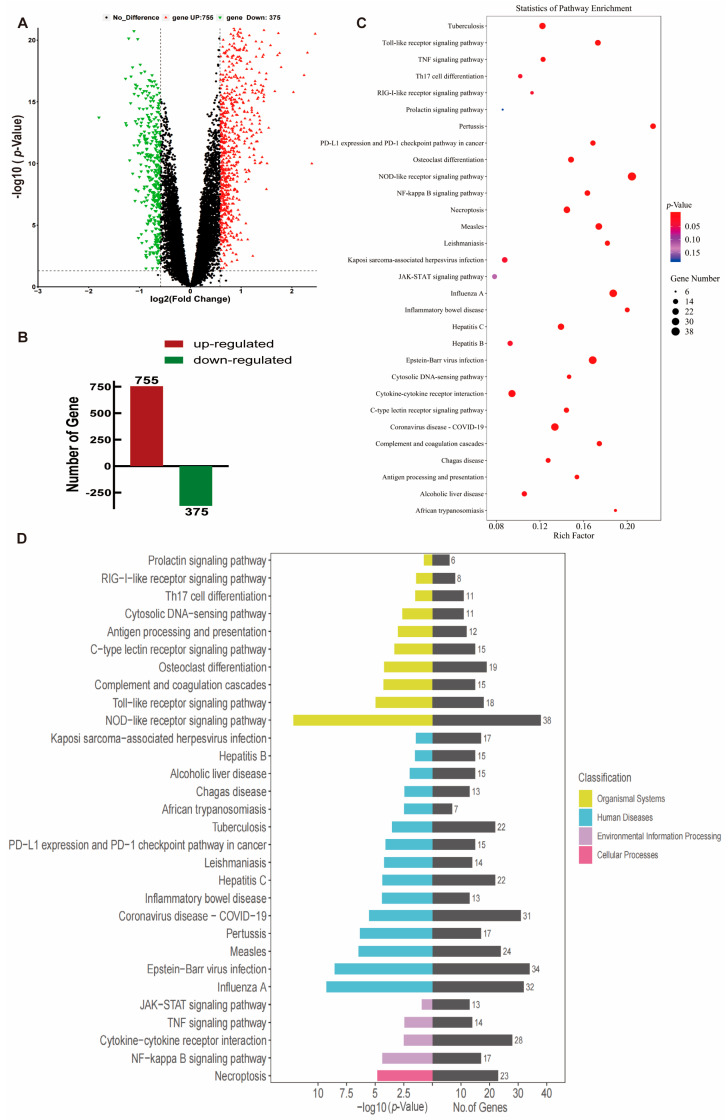
Volcano Plots and KEGG pathway enrichment analysis results of DEGs in peripheral blood of PTB patients and healthy people. Bubble and Bar diagrams show the top 30. (**A**) Volcano plots of DEGs. (**B**) Histograms of Gene number. (**C**) KEGG bubble diagram. (**D**) KEGG bar diagram. FC ≥ 1.5, *p* < 0.05.

**Figure 3 ijms-24-12709-f003:**
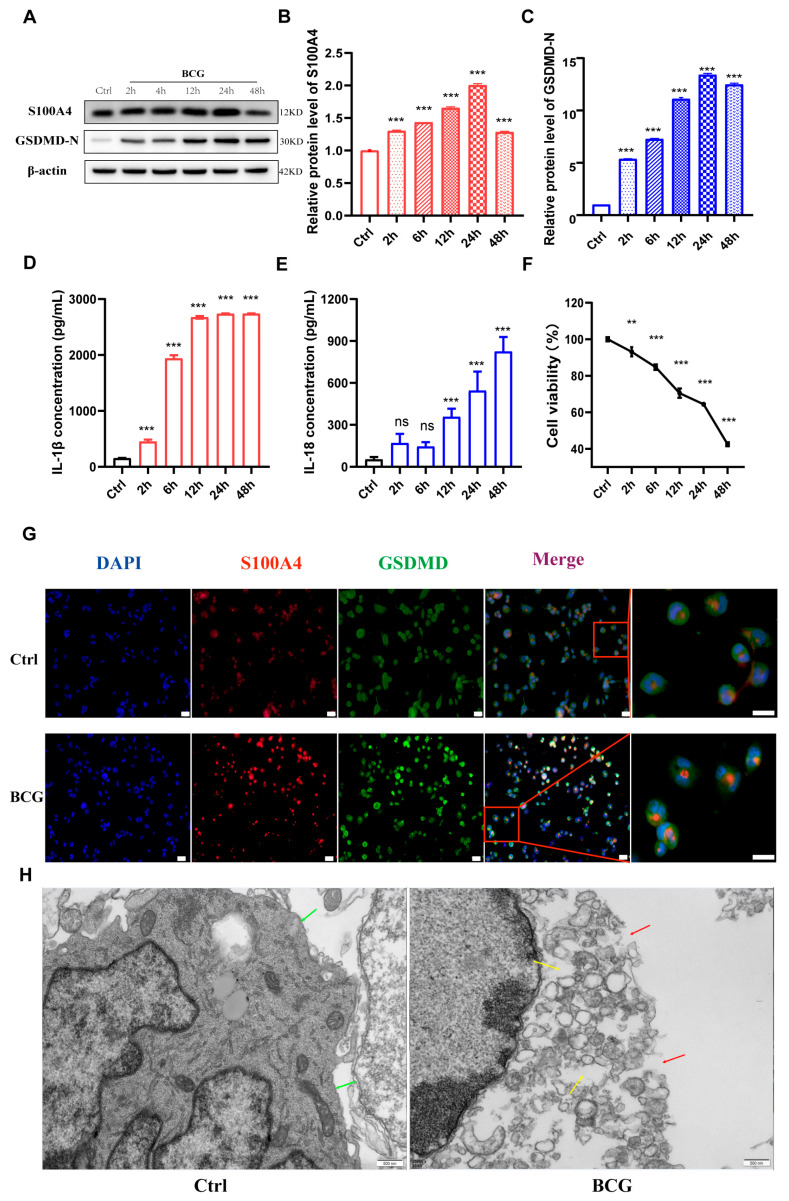
BCG infection up-regulated expression of *S100A4* and induced pyroptosis in THP-1 macrophage. (**A**–**C**) *S100A4* and GSDMD protein levels were measured at indicated time points in BCG-infected THP-1 macrophages (MOI = 10). (**D**,**E**) Concentrations of IL-1β and IL-18 in the supernatants of BCG-infected THP-1 macrophages were measured using ELISA at the indicated time points. (**F**) CCK-8 was used to detect cell viability at the specified times. (**G**) *S100A4* and GSDMD proteins were quantified using immunofluorescence staining after 24 h of BCG infection. Blue, red, and green spots correspond to DAPI, *S100A4*, and GSDMD, respectively. Scale bar: 20 μm. (**H**) The blank control group (Ctrl) and BCG treatment group (BCG) were observed using a transmission electron microscope 24 h after BCG infection. Green arrows point to intact cell membranes, red arrows point to membrane perforations, and yellow arrows point to lipid droplets. Scale bar: 500 nm. Data are from mean ± SD of three independent experiments, ^ns^ *p* > 0.05, ** *p* < 0.01, *** *p* < 0.001.

**Figure 4 ijms-24-12709-f004:**
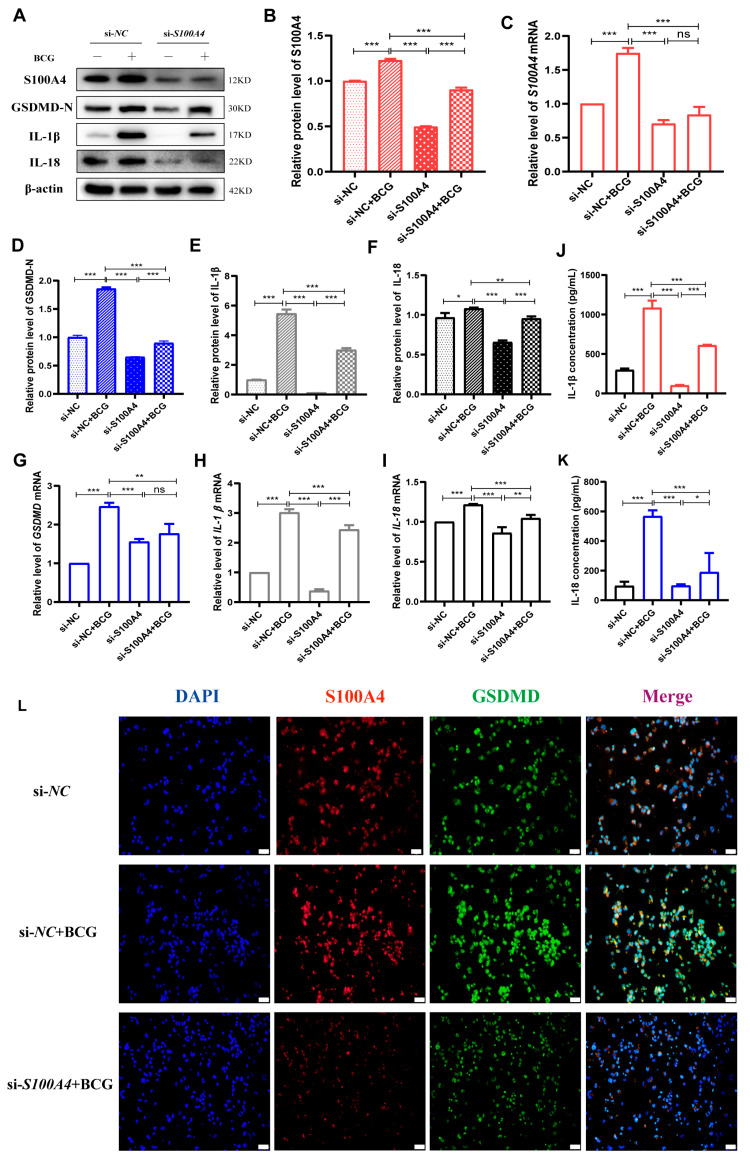
Knockdown of *S100A4* inhibited pyroptosis in BCG-infected THP-1 macrophage. (**A**–**C**) The knockdown efficiency of si-*S100A4* was detected using Western blots and qRT-PCR. (**D**–**F**) The levels of pyroptosis-related proteins (GSDMD-N, IL-1β p17, IL-18 p22) were detected using Western blots in cells pre-treated with si-*S100A4* for 24 h and infected with BCG for 24 h. (**G**–**I**) qRT-PCR was used to detect the expression of pyroptosis-related mRNA in THP-1 macrophages pre-treated with si-*S100A4* for 24 h and infected with BCG for 2 h subsequently. (**J**,**K**) ELISA was used to detect the concentrations of IL-1β and IL-18 in the supernatant of cells pre-treated with si-*S100A4* for 24 h and infected with BCG for 24 h. (**L**) Immunofluorescence staining was used to compare the morphological changes of *S100A4* and GSDMD cells in si-*NC*, si-*NC*+BCG, and si-*S100A4*+BCG groups. Blue, red, and green spots correspond to DAPI, *S100A4*, and GSDMD, respectively. Scale bar: 20 μm. Data are from mean ± SD of three independent experiments, ^ns^
*p* > 0.05, * *p* < 0.05, ** *p* < 0.01, *** *p* < 0.001.

**Figure 5 ijms-24-12709-f005:**
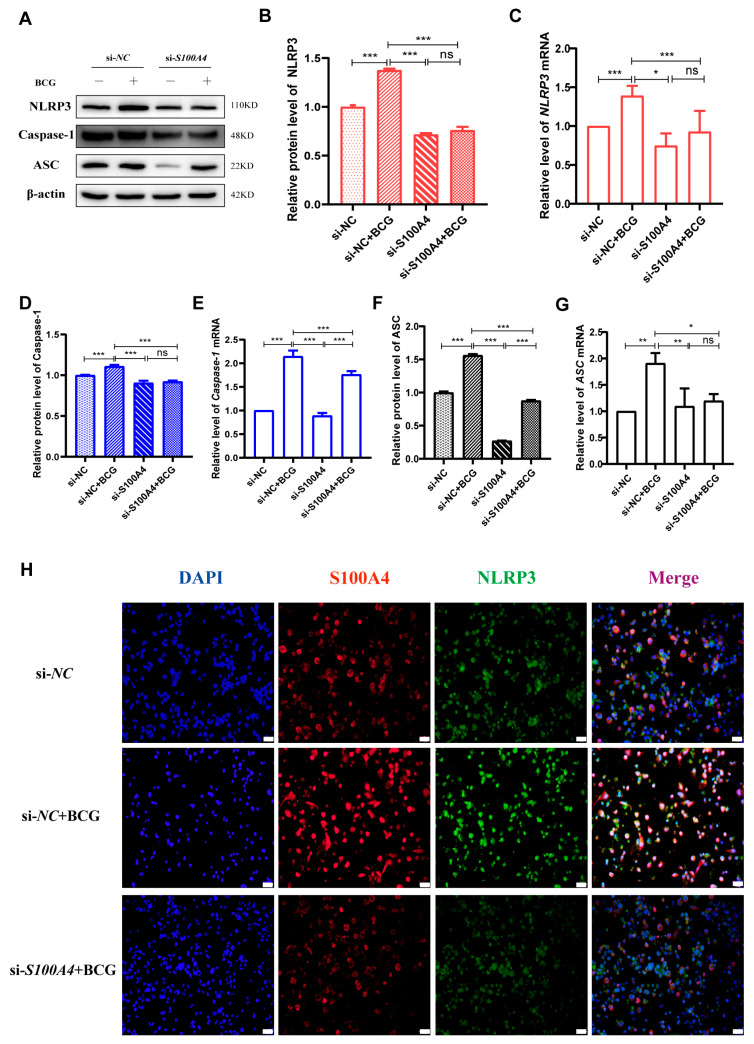
*S100A4* knockdown down-regulated inflammasome-related proteins in BCG-infected THP-1 macrophages. (**A**–**G**) Western blot and qRT-PCR were used to detect the expression of inflammasome-related proteins (NLRP3, Caspase-1 p48, ASC) and mRNA in macrophages pre-treated with si-*S100A4* for 24 h, followed by BCG infection for another 24 h. (**H**) Immunofluorescence staining was used to compare the morphological changes of *S100A4* and NLRP3 in si-*NC*, si-*NC*+BCG, and si-*S100A4*+BCG groups. Blue, red, and green spots correspond to DAPI, *S100A4*, and NLRP3, respectively. Scale bar: 20 μm. Data are from mean ± SD of three independent experiments, ^ns^ *p* > 0.05, * *p* < 0.05, ** *p* < 0.01, *** *p* < 0.001.

**Figure 6 ijms-24-12709-f006:**
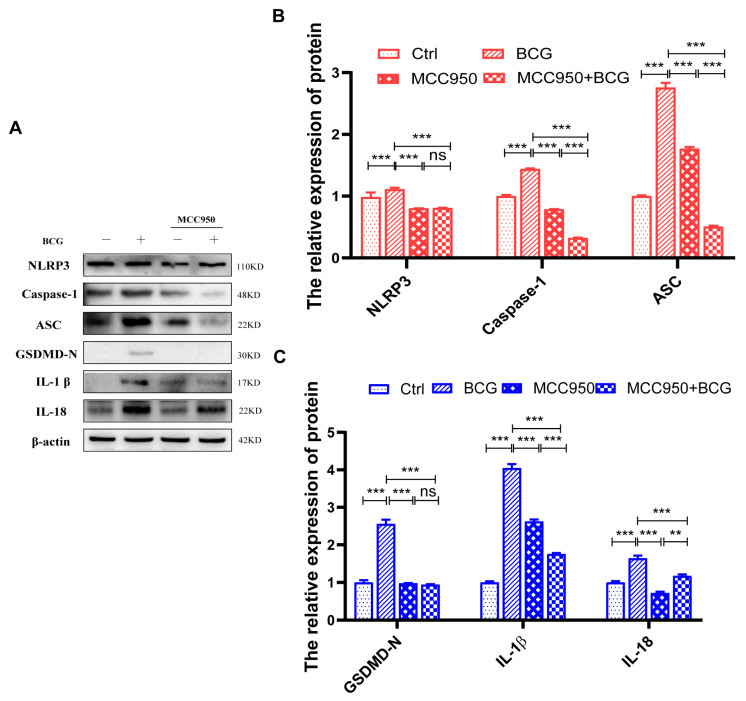
Inhibition of NLRP3 resulted in down-regulation of the expression of inflammasomes and pyroptosis-related proteins. THP-1 macrophages were pre-treated with NLRP3 inhibitor MCC950 10 μM for 2 h, followed by BCG infection for 24 h (**A**,**B**). Compared to the BCG group, the expression of the inflammasome-related proteins (NLRP3, Caspase-1 p48, ASC) was downregulated in the MCC950+BCG group. (**A**,**C**) Compared to the BCG group, the expression of the pyroptosis-related proteins (GSDMD-N, IL-1 p17, IL-18 p22) was down-regulated in the MCC950+BCG group. Data are from mean ± SD of three independent experiments, ^ns^
*p* > 0.05, ** *p* < 0.01, *** *p* < 0.001.

**Figure 7 ijms-24-12709-f007:**
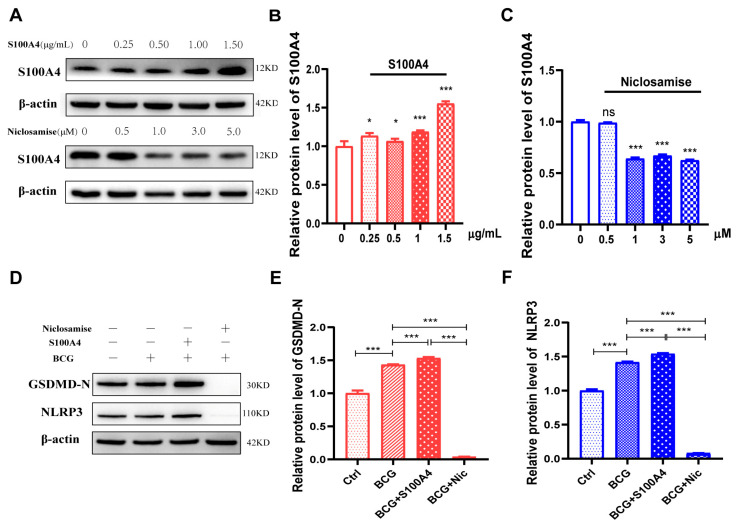
Exogenous *S100A4* up-regulated the protein expression of GSDMD-N/NLRP3, while Nic down-regulated them. (**A**–**C**) THP-1 macrophages were pre-treated with exogenous *S100A4* or *S100A4* inhibitor Nic for 2 h and then infected with BCG for 24 h. The expression of *S100A4* was detected using Western blot to determine the optimal concentration. (**D**–**F**) THP-1 macrophages were pre-treated with 1.5 ug/mL exogenous *S100A4* and 1 uM Nic for 2 h, respectively, and then infected with BCG for 24 h. The expression of related proteins (GSDMD-N and NLRP3) was detected using Western Blot. Data are from mean ± SD of three independent experiments, ^ns^ *p* > 0.05, * *p* < 0.05, *** *p* < 0.001.

**Figure 8 ijms-24-12709-f008:**
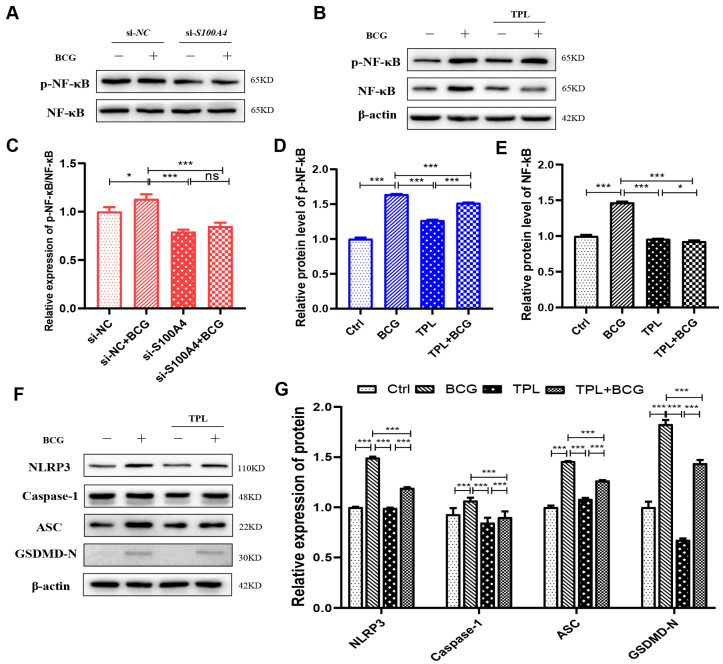
Knockdown of *S100A4* inhibited the NF-κB signaling pathway and reduced the expression of NLRP3 inflammasome and pyroptosis-related proteins. (**A**,**B**) si-*S100A4* downregulated the expression of p-NF-κB. The protein expression levels of p-NF-κB and NF-κB were analyzed using Western blot in the THP-1 macrophages pre-treated with si-*S100A4* for 24 h and infected with BCG for 24 h subsequently. (**C**–**E**) The protein expression levels of p-NF-κB and NF-κB in the THP-1 macrophages, which were pre-treated with Triptolide (TPL) 5 nM for 2 h and infected with BCG for 24 h, were quantitatively analyzed using Western blot. (**F**,**G**) Inhibition of NF-κB decreased the expression of inflammatory-related proteins (NLRP3, Caspase-1 p48, ASC) and a pyroptosis-related protein (GSDMD-N). Data are from mean ± SD of three independent experiments, ^ns^
*p* > 0.05, * *p* < 0.05, *** *p* < 0.001.

**Figure 9 ijms-24-12709-f009:**
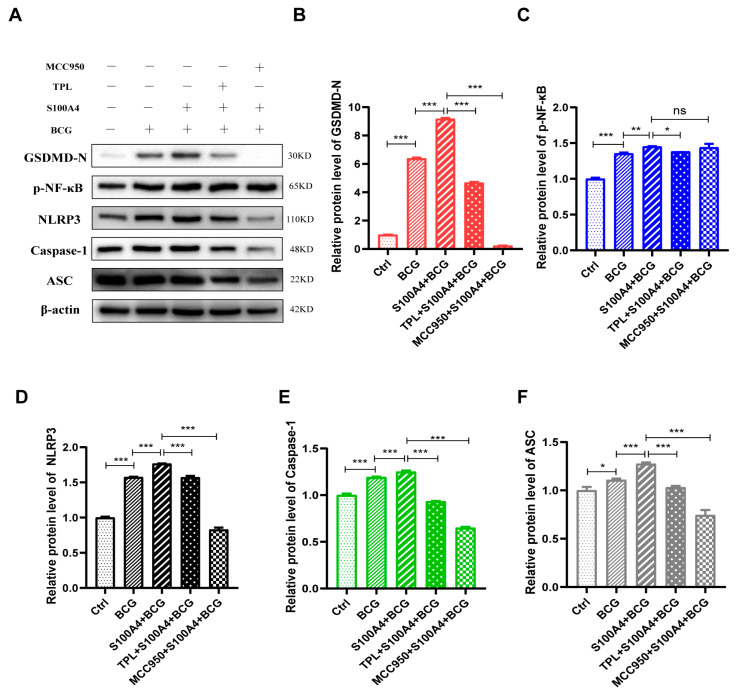
*S100A4* activates the NF-κB/NLRP3 inflammasome signaling pathway to promote pyroptosis. THP-1 macrophages were pre-treated with MCC950 and TPL for 2 h, followed by the addition of exogenous *S100A4*, and then infected with BCG for 24 h. (**A**–**F**) Western blot was used to detect the protein expression of (**B**) GSDMD-N, (**C**) p-NF-κB, (**D**) NLRP3, (**E**) Caspase-1, and (**F**) ASC. Data are from mean ± SD of three independent experiments, ^ns^ *p* > 0.05, * *p* < 0.05, ** *p* < 0.01, *** *p* < 0.001.

**Figure 10 ijms-24-12709-f010:**
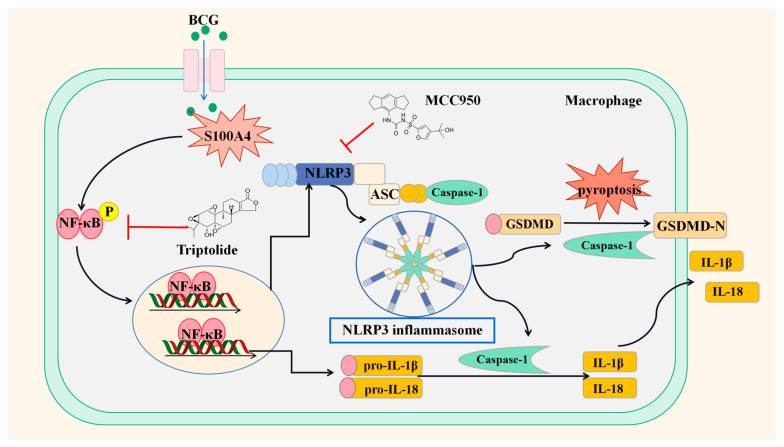
Schematic diagram of *S100A4* promoting BCG-induced pyroptosis in THP-1 macrophages through activation of the NF-κB/NLRP3 inflammasome signaling pathway.

**Table 1 ijms-24-12709-t001:** Summary of sample information of GSE83456.

Item	Content
Title	The transcriptional signature of active tuberculosis reflects symptom status in extrapulmonary and pulmonary tuberculosis
Organization name	National Institute for Medical Research (NIMR)
Status	Public on 4 November 2016
Organism	Homo sapiens
Sample group	61 healthy human controls, 47 humans with EPTB, 45 humans with PTB
Sample type	Peripheral blood mRNA transcriptome
Platforms	GPL10558 Illumina HumanHT-12 V4.0 expression bead chip

**Table 2 ijms-24-12709-t002:** Sequences of small interfering RNAs to *S100A4* genes.

Gene Name	si*RNA* Sequences of *S100A4*
Nontarget control (si-*NC*)	Sense: 5′-UUCUCCGAACGUGUCACGUTT-3′ Anti-sense:3′-ACGUGACACGUUCGGAGAATT-5′
si-*S100A4*	Sense: 5′-UCCAGAAGCUGAUGAGCAATT-3′ Anti-sense:3′-UUGCUCAUCAGCUUCUGGATT-5′

**Table 3 ijms-24-12709-t003:** Primers information of qRT-PCR.

Gene	Primer Sequence (5′-3′)	Tm (°C)	Product Size (bp)
β-actin	F: CCTGGCACCCAGCACAAT	60	144
	R: GGGCCGGACTCGTCATAC	59	
*S100A4*	F: CGGGCAAAGAGGGTGACAAGTTC	64	145
	R: TTGTCCCTGTTGCTGTCCAAGTTG	64	145
GSDMD	F: TGGACCCTAACACCTGGCAGAC	64	117
	R: GCACCTCAGTCACCACGTACAC	63	117
NLRP3	F: CTCGGTGACTTCGGAATCAGACTTC	63	144
	R: CAGGGAATGGCTGGTGCTCAATAC	64	144
ASC	F: TGGATGCTCTGTACGGGAAGGTC	64	131
	R: CAAGTCCTTGCAGGTCCAGTTCC	63	131
Caspase-1	F: GGTGCTGAACAAGGAAGAGATGGAG	63	104
	R: CCTGTGCCCCTTTCGGAATAACG	64	104
IL-18	F: TGACCAAGGAAATCGGCCTC	60	116
	R: CCATACCTCTAGGCTGGCTATCT	60	116
IL-1β	F: TACGAATCTCCGACCACCACTACAG	64	137
	R: GGGAAAGAAGGTGCTCAGGTCATTC	64	137

## Data Availability

The original contributions presented in the study are included in the Article.

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
