# Peer review of "S100A4 Promotes BCG-Induced Pyroptosis of Macrophages by Activating the NF-κB/NLRP3 Inflammasome Signaling Pathway"

_ijms, 2023, doi:10.3390/ijms241612709_

Round 1

Reviewer 1 Report

This study by Li et al., presented the data on macrophage pyroptosis induced by BCG via NF-κB-NLRP3 inflammasome. Authors utilized publicly available tuberculosis patinets and healthy individuals perpheral blood mRNA transcriptome data, and found that S100A4 and GSDMD were significantly up regulated in active pulmonary tuberculosis patients. Authors validated these findings, authors infected THP-1 macrophages with BCG and analyzed S100A4 and GSDMD. Manuscript writing and data presentation are good, but could be improved. The below are my concerns, which may authors consider to improve the manuscript. 

1. I am not in support of presenting the data analysis from public data base in to the main figures. Either, these figures (figure 1, 2) should be moved to supplementary or if not, at least I would recommend to mention the source like ' Blankley et al., 2016' in the figure.

2. Figure 3-G: Authors should provide the IF images with macrophage marker

Scale bar is not readable. 

3. Figure 4: IF images must be supplied with macrophage marker staining.

4. Figure 5. IF images must be provided with macrophage marker staining.

Reviewer 2 Report

The manuscript describing the role of S100A4 in pyroptosis in the context of MTb is well written and the conclusions are supported by the results provided. There are following concerns.

1. data mining analysis to explore the correlation between S100A4, GSDMD and PTB progression- This suggest that the authors were predetermined and focused to explore the role of S100A4 in the data. Data mining and analysis should be done without bias and if S100A4 and GSDMD appears on the list of DEGs, then only it should be reported. Please reformat this sentence.

2.How many DEGs were listed out? What were the criteria used (Log FC and p value) before running out analysis for DEGs?

3. While conducting KEGG, please list the number of genes for each pathway along with p values. Figure 2 panel A suggest that number of genes are much less in TN compared to other disease and TB is at 11th position. Does S100A4 was listed in this category of DEGs. Please provide the list of DEGs involved in first 10 or at least in TB.

4 Bio-informatics section should be in more detail.

5. western blot should be Western Blot.

6. figure 4 panels B-K, Figure 5 panels B-G, Figure 6 panels B and C, Figure 7 panels B, C, E, and F, Figure 8 panels C-E and G- statistical analysis should be done in between all groups as done in Figure 9.

7.Figure 6 panel A- please provide a new blot for IL-1B and IL-18.

8. Figure 3 panel A- why there was a decrease in S100A4 at 48 hours?

9. Discussion is written like a review article, please correlate the results of this study with existing literature.

Round 2

Reviewer 1 Report

I appreciate authors for improving the MS. However, I am not convinced by CD11b staining for monocytes/macrophages. Therefore, my concern remains the same.

Author Response

Thank you again for your positive comments and valuable suggestions to improve the quality of our manuscript.

Reviewer 2 Report

None

Author Response

(The authors gave the same response as above.)
